# Clinical Evaluation of Dexamethasone Plus Gentamycin Mouthwash Use in Combination with Total Glucosides of Paeony for Treatment of Oral Lichen Planus without Fungal Infection: A Comparative Study with Long-Term Follow-Up

**DOI:** 10.3390/jcm11237004

**Published:** 2022-11-27

**Authors:** Zhihui Zhang, Yan Jia, Liyuan Tao, Xiaodan Liu, Ying Han, Xiao Wang

**Affiliations:** 1Department of Stomatology, Peking University Third Hospital, Beijing 100191, China; 2Research Center of Clinical Epidemiology, Peking University Third Hospital, Beijing 100191, China; 3Department of Oral Medicine, Peking University School and Hospital of Stomatology, Beijing 100081, China

**Keywords:** oral diseases, oral lichen planus, dexamethasone, gentamicin, total glucosides of paeony, combined treatment

## Abstract

Background: Oral lichen planus (OLP) is a common chronic inflammatory disease of the oral mucosa and considered a potential malignant disease, for which a method for complete cure is lacking. The dexamethasone and gentamicin mouthwash, combined with total glucosides of paeony (TGP), was tested in the treatment of OLP patients without fungal infection, with a view to provide evidence that may assist in resolving the dilemma. Methods: A randomized and single-blind clinical trial of 48 non-erosive and erosive OLP patients was conducted, with the patients divided into two groups. Group A was treated with dexamethasone plus gentamycin mouthwash and Group B received an additional TGP capsule together with the aforementioned mouthwash. All the patients were followed up with four times, at 2 weeks, 1 month, 3 months, and 6 months. The clinical manifestations, sign score, and VAS scale were recorded. The total effective rate (%) was defined as (cases of complete resolution + cases of partial resolution)/total cases observed × 100%. Results: A total of 43 patients completed all follow-up appointments. Among the 21 patients in Group A, the total effective rate was 61.9%. Of the 22 patients in Group B, the total effective rate was 89.66%. The clinical manifestation, sign score, and VAS of the two groups all indicated improvements, and there were significant differences between the two groups (*p* < 0.05). Conclusions: Dexamethasone plus gentamycin mouthwash combined with TGP treatment for OLP patients is a safe and effective treatment of OLP.

## 1. Introduction

Lichen planus is a chronic inflammatory disease that affects the skin, hair follicles, nails, and mucous membranes. Oral lichen planus (OLP) is a mucosal subtype of lichen planus (LP), the prevalence of which is about 0.5–2%, and middle-aged women are the population most prone to the disease. OLP was described for the first time in 1866, with an unclear etiology or pathogenesis and uncertain premalignant potential [1]. With the developments in research in recent years, the etiology of OLP has gradually come to be considered as being closely related to immune dysfunction, psychoneurologic factors, genetic factors, and infection factors, among others [2]. OLP has been classified as one of the common oral potentially malignant disorders (OPMDs) in clinical practice [3], with a malignant transformation rate ranging from 0 to 3.5% [4]. OLP clinical manifestations, including reticulate features on the single or more sites of oral mucosal as the main type, which may be accompanied by hyperemia or erosion.

At present, there is no radical cure for OLP, and scholars have focused on finding treatment methods, including topical and systemic pharmacotherapy, to improve the clinical therapeutic effect. First-line treatments, described in the guidelines on the management of LP [5], include topical steroids, intralesional injection of corticosteroids, systemic corticosteroids, and retinoids such as acitretin initially followed by isotretinoin [6]. Furthermore, several approaches were considered for OLP treatment, including water and glycerol mouth rinses, pastes and mucoadhesive polymers, and the most promising therapeutic effect observed was a mixed mouth rinse containing clobetasol. By comparison, dexamethasone was not inferior to clobetasol combined with ketoconazole/amitriptyline as an effective strategy for OLP [7]. Therefore, topical application of dexamethasone, alongside other ultrapotent steroids, is the mainstay of localized OLP treatment and is proven to be safe and effective [5,8].

Gentamicin is a broad-spectrum antibiotic with anti-inflammatory and antibacterial effects. It can be combined with clindamycin as a feasible treatment protocol for pelvic inflammatory disease [9]. Gentamicin exhibits bactericidal activity and synergizes with lithocholic acid to achieve an antibiofilm effect against *L. monocytogenes*, which leads to bacterial death [10]. In the previous study, a local application of compound gentamicin gargle, configured with dexamethasone sodium phosphate injection and gentamycin sulfate injection in 0.9% sodium chloride injection solution, exerted a therapeutic effect on erosive OLP [11]. Accordingly, with their facility of configuration and operability, local application of gentamicin in combination with dexamethasone may be a promising approach for the inflammation of OLP.

Total glucosides of paeony (TGP) are extracted from the root and stem of paeony, with the composition of paeoniflorin, hydroxyl paeoniflorin, paeonin, albiflorin, and benzoylpaeoniflorin, and more than 90% is paeoniflorin (Pae) [12]. In cases recalcitrant to topical corticosteroids, the application of azathioprine [13] and methotrexate [14] can be suggested as the second-line treatments for OLP. However, immunosuppressants such as methotrexate and hydroxychloroquine, which suppress immune cells, may be limited in their application by the presence of side effects, such as infections. TGP decreased inflammatory substance production via anti-inflammatory and immunoregulatory effects in synoviocytes in rheumatoid arthritis patients [15]. TGP effectively alleviates the symptoms of DSS-induced colitis [16]. OLP patients receiving TGP can experience significant improvements in clinical symptoms with fewer side effects [17,18].

We hypothesized that dexamethasone plus gentamicin mouthwash in combination with TGP capsules, is more effective compared with this mouthwash alone on the treatment of OLP. Therefore, the objective of our study was designed to evaluate the efficacy and safety of dexamethasone plus gentamicin mouthwash, or its combination with TGP capsules on OLP patients without fungal infection.

## 2. Materials and Methods

### 2.1. Patients

This study was a prospective, single-blind, randomized controlled clinical trial between September 2020 and February 2022 at the Department of Stomatology, Peking University Third Hospitals in Beijing, China. Using SAS 9.4 software (SAS Inc., Cary, NC, USA), the subjects were randomly divided into two groups: A and B. Patients who were enrolled in the study met the following inclusion criteria: (1) had a diagnosis of oral lichen planus both clinically and histopathologically; (2) were aged over 18 years; (3) had not received local or systemic corticosteroids or other immunomodulatory drugs within 3 months before onset; (4) were without serious dental and periodontal diseases; (5) had no fungal infection; (6) were willingness to participate in the study and gave informed consent. Subjects were excluded from our study if they were: (1) aged under 18 years; (2) pregnant or lactating; (3) had serious systemic diseases of the heart, lung, liver, and kidney, or had tumors or mental illness; (4) were taking other regular medication, which could possibly cause drug-induced OLP; (5) were unwilling to participate in the experiment. The study was approved by the Ethics Committee of the Third Hospital of Peking University (M2020107). All the participants signed an informed consent before the trial started. Patients who met the inclusion criteria in the outpatient clinic and were willing to participate in the study after informed consent were consecutively enrolled in the study to ensure that the overall patients were well represented. During the study, patients could withdraw at any time if they were uncomfortable or not willing to continue the treatment.

### 2.2. Treatment Protocol

Patients in Group A were topically treated with dexamethasone plus gentamicin mouthwash as the control group. Dexamethasone sodium phosphate injection 5 mg/mL/bottle × 2 (Sinopharm Rongsheng Pharmaceutical Co., Ltd., Sinopharm H41020036, Henan, China) and gentamycin sulfate injection 80 mg/2 mL/bottle × 2 (Yichang Renfu Pharmaceutical Co., Ltd., Sinopharm H42022058, Yichang, China) were each mixed separately in a 500 mL of 0.9% sodium chloride injection solution (Beijing Shuanghe Pharmaceutical Co., Ltd., Sinopharm H11021192, Beijing, China), which was applied twice daily for 3 weeks, with a 1 week pause, and continued in this manner until symptoms subsided. The experimental Group B patients received total glucosides of paeony capsules (Ningbo Lihua Pharmaceutical Co., Ltd., H20055058, Ningbo, China), 2 capsules (0.3 g/capsule) thrice daily, together with the above mouthwash. In order to ensure the accuracy and reliability of records, the information of all patients’ initial and follow-up visits was controlled by the same fixed oral medicine specialist who conducted registration, took photos, and saved records to the medical record report form, ensuring detailed inclusion and analysis of standardized and unified data. In addition, the drugs were from the same source. The dexamethasone and gentamicin mouthwash were dispensed by professional nurses. An amount of 5 mL was gargled for 5 min and expectorated, then eating or drinking was avoided for 15 min after rinsing, according to the previous means of mouthwash [19]. All patients were given oral health guidance and advised to maintain oral hygiene, including effective brushing (BASS brushing method) and regular flossing. They were followed up with four times, at 2 weeks and at 1, 3, and 6 months after treatment. The degree of signs, symptoms, and clinical manifestations were recorded at each visit. Compliance control was adopted by the method of making an appointment for the next visit every time, and making a phone call for confirmation of the patients the day before the follow-up.

### 2.3. Clinical Recording

All patients underwent a clinical evaluation at baseline (Day 0) and at 2 weeks and 1, 3, and 6 months for follow-up. The patient’s symptoms, symptom score, and sign score were recorded on the case report form for color and changes in lesion area, which consisted of a diagram of the oral mucosa with hyperkeratotic, atrophic, or erosive lesions at each visit, according to the criteria [20]. The sign score data were scored according to a previously used criteria scale: 5 = white striae with erosive area > 1 cm^2^; 4 = white striae with erosive area < 1 cm^2^; 3 = white striae with atrophic area > 1 cm^2^; 2 = white striae with atrophic area < 1 cm^2^; 1 = mild white striae without erythematous or erosive area; 0 = no lesions. The symptom scores were measured on a visual analogue scale (VAS) and divided into 10 grades: 0 = no pain (VAS: 0); 1 = mild pain (VAS: 1–3), 2 = moderate pain (VAS: 4–6); 3 = severe pain (VAS: 7–10) according to previous studies [18,21,22]. A complete resolution of the clinical symptoms was defined as the absence of all atrophic-erosive lesions or lack of white striae and a VAS value of 0. Partial resolution was defined as a reduction of the lesion area compared with the original lesion or if the color became lighter and if the VAS value decreased. No resolution was defined as no change in the original lesion area, if the area expanded, or if the VAS value increased. The primary outcome was clinical response (sign score) and the secondary outcome was symptom score (VAS). The evaluator of primary outcome was a different oral medicine specialist than the person who administered the treatment. She did not participate in the enrollment phase of the patients and only evaluated the outcome measures for blinding ensured. VAS was completed by the patients and recorded by the doctor. During the clinical examination, the presence of clinical manifestations of oral candidiasis, such as pseudomembranous, erythema, or proliferative candidiasis, angular cheilitis, or denture stomatitis was noted. The total effective rate (%) was defined as (cases of complete resolution + cases of partial resolution)/total cases observed × 100%.

Laboratory tests, including routine blood examination and hepatic and renal function, were performed at baseline and at the second, third, and fourth follow-up visit. At the same time, a cytological smear was performed, and was positive if it showed at least one candida filament or pseudohyphal cell. Fungal culture and microbial identification tests were used to identify the species. Subjects were checked for side effects during the therapeutic session in the groups.

The sample size was calculated based on the percentage of lesion area reduction according to a previous study [19]. The mean lesion area was reduced by 54% ± 12.5% in the control group and 65% ± 12.5% in the treatment group. The sample size ratio of the experimental group and the control group was 1:1, and *α* = 0.05 and *β* = 0.2 were set for hypothesis testing according to the formula for calculating the sample size.

$n_{2}=\frac{{\left(z_{1-\alpha /2}+z_{1-\beta }\right)}^{2}\left(sd_{1}^{2}+sd_{2}^{2}\right)\left(1+1/k\right)}{2{\left(mean_{1}-mean_{2}\right)}^{2}}$, $n_{1}=k\times n_{2}$, $z_{1-\alpha /2}$ = 1.96, $z_{1-\beta }=$ 0.84, resulting in a sample size of 21 for the experimental and control group. Assuming a loss-to-follow rate of 10%, a minimum sample size of 24 each was required for the experimental group and control group.

### 2.4. Statistical Analysis

According to the principle of intention-to-treat (ITT) analysis, this study used three datasets (mITT,PP,SS) to statistically analyze the results. Among them, mITT set was defined as: patients who participated in the second follow-up and obtained the results after randomization of the study objects can enter the data set. The PP set was defined as patients who completed 4 follow-up visits after randomization to enter the dataset. The SS set was defined as all patients taking medication who were recorded after randomization of the study subjects.

The statistical analyses were performed using SPSS 26.0 (IBMCorp., Armonk, NY, USA). The data are expressed as means ± standard error of mean (SEM). Chi-square analysis, and one-way ANOVA test were used to compare the responses between the groups. Two-sided *p* < 0.05 was considered to indicate a statistically significant difference.

## 3. Results

Of the 48 patients, 43 completed the follow-up, including 21 patients in Group A and 22 patients in Group B. Two dropped out due to incomplete results (one case in Group A and one case in Group B), who only participated in the first visit and did not come back for follow-up visit, two dropped out due to missed follow-up in Group A for personal reasons, and one dropped out of the study due to intolerable diarrhea in Group B, as shown in Figure 1. In this study, the research objects of modified ITT (mIITT) and PP were the same. The area most often affected by lesions was the buccal mucosa, either as a single site or accompanied by lesions in the gingiva, tongue, lip, palate, and other sites. There were no statistically significant differences in the gender, age, type, location, and other basic information between the patients of Groups A and B (Table 1).

In the modified mITT set, baseline symptoms were recorded in all patients at the time of initial diagnosis. There were no significant differences in pretreatment sign score and in symptom score VAS between OLP patients in each group. During the follow-up period, the sign scores and VAS of both groups decreased significantly (Table 2). From the second week to the sixth month after treatment, the signs score of the two groups gradually changed and further decreased at the third month, thereafter remaining stable to the sixth month. A better effect was obtained in Group B, and the difference was statistically significant (*p* < 0.05, Figure 2A). The VAS and sign changes in Group B were better than those in Group A (*p* < 0.05, Table 2). In the first month, the VAS of Group B showed a significant decrease and remained stable up to the sixth month (Figure 2B).

Treatment outcomes were assessed in both groups after 2 weeks and 1, 3, and 6 months of treatment (*p* < 0.05, Table 3). During the fourth follow-up visit at 6 months, there were 21 cases in total; 1 case showed complete resolution and 12 cases partial resolution, showing an effective rate of 61.9% in Group A. Meanwhile, there were 22 cases in total, 4 cases of complete resolutions, and 15 cases of partial resolution, showing an effective rate of 86.4% in Group B. The efficacy of Group B was significantly higher than that of Group A during the follow-up course (45.5% vs. 33.3%, 68.2% vs. 52.4%, 81.8% vs. 66.7%, and 86.4% vs. 61.9%).

For a single site of mucosal involvement, i.e., buccal, gingiva, or palate alone, there was no difference in sign score or VAS between the two groups at baseline and the follow-up after treatment (Table 4). The change trend in each group is shown in Figure 2C,D. For more than one site of mucosal involvement, there was no significant difference in sign score and VAS between the two groups at baseline. At 3 months and 6 months, the sign score in Group B showed significantly greater improvements than that in Group A (*p* < 0.01, Table 4), and the VAS showed significantly lower symptom severity for Group B than Group A at the 1-, 3-, and 6-month follow-up (*p* < 0.01), as shown in Table 4. The change trend in each group can be seen in Figure 2E,F. For a single site of mucosal involvement, there was no significant difference in the effective rate between Group A and Group B at 6 months follow-up (71.4% vs. 71.4%), as shown in Table 5. For more than one site of mucosal involvement, the effective rate of Group B was better than that of Group A (93.3% vs. 57.1%), as shown in Table 5 (*p* < 0.01).

There was no significant difference in pretreatment sign scores and VAS between the groups with non-erosive OLP. From 2 weeks to 6 months after treatment, sign scores changed gradually in both groups, with no significant recurrence in either Group A or B, and there was no significant difference between groups A and B (Figure 3A). In terms of pain relief, VAS in Group B showed improvements at 1 month and significant relief at 6 months, while VAS in Group A indicated relief at 3 months, which was maintained at 6 months. The degree of VAS relief in Group B was more obvious than that in Group A, but the difference was not significant (Figure 3B). Two clinical examples of Groups A and B with non-erosive OLP are shown in Figure 4. The treatment outcomes were assessed in both groups with non-erosive OLP at 2 weeks and 1, 3, and 6 months of treatment (Table 6). There was no difference in the effective rate between the two groups (50.0% vs. 44.4%, 70.0 vs. 77.8%, 80.0% vs. 88.9%, and 80.0% vs. 88.9%).

There was no significant difference at baseline between erosive OLP patients in both groups. From 2 weeks to 6 months after treatment, the signs of the two groups gradually changed. From 1 month, the degree of sign score improvement was significantly more pronounced for Group B than for Group A. The sign score of Group B, which was maintained at 6 months, was significantly better than that of Group A (Figure 5A). The VAS of both groups gradually decreased from 2 weeks to 6 months after treatment. From 3 months until 6 months, the VAS indicated significantly greater relief for Group B than for Group A, after which there was partial recurrence in Group A but no recurrence in Group B (Figure 5B). Two clinical examples of Groups A and B with erosive OLP are shown in Figure 6. Patients with erosive OLP in both groups were evaluated for efficacy after 2 weeks and 1, 3, and 6 months of treatment (Table 6). The effective rate of Group B was significantly higher than that of Group A (46.2% vs. 18.2%, 61.5% vs. 36.4%, 76.9% vs. 54.5%, and 84.6% vs. 45.5%).

In the safety set (SS), all patients (*n* = 48) in the study showed no abnormalities according to routine blood tests and kidney function tests. The number of adverse events (AEs) in both groups is reported in Figure 1 and Table 6. Side effects in the oral cavity included candidiasis, one patient in Groups A and B separately for erosive OLP, and in less than 5% of cases in both groups. Experimental drugs were withdrawn due to gastrointestinal symptoms (diarrhea) in one patient of Group B. Systemic side effects in Group B occurred in one patient, with abnormal liver function (total bilirubin) identified at the fourth follow-up. There was no significant difference in both groups (*p* > 0.05, Table 7).

## 4. Discussion

In our study, the control group showed a significant improvement in the condition of patients affected by oral lichen planus (OLP) treated with dexamethasone plus gentamicin mouthwash. Its combination with total glucosides of paeony (TGP) capsules achieved good results vs. the control group. For non-erosive OLP, the effect indicated gradual improvements in both groups with no significant difference. The advantage was more pronounced for erosive OLP patients, and the effect for combination therapy was significantly better. Clinically, mouthwash alone was effective for single site mucosal involvement, while for the larger scale OLP with more than one site of mucosal involvement, TGP in combination with the mouthwash will significantly increase the efficacy.

OLP has generated intense discussion and been more controversial than any other disease in oral pathology and medicine in many years [1]. As the background of OLP has not been fully explained, the current therapies are largely symptom-directed and focus on relieving symptoms [3], with monitoring and preventing carcinogenesis to a certain extent, but a permanent cure for OLP it is not yet possible. The typical pathological features of OLP are liquefaction of the mucosal epithelium basal layer and dense infiltration of T lymphocytes in the adjacent subepithelial lamina propria [4]. There are both characteristic disease activities and individual differences in OLP, which makes it difficult to achieve a satisfactory therapeutic effect with a single specific treatment strategy [3]. The challenge for any combination of multiple drugs, especially local and systemic immunomodulatory ones, is to promote and improve their therapeutic effect.

Among the available therapeutic options, corticosteroids are considered the primary treatment due to their rapid effect in controlling OLP symptoms [5]. Drugs such as dexamethasone, clobetasol propionate, triamcinolone, betamethasone, fluocinonide, prednisolone, and fluticasone can be applied topically, either in mouthwashes, orabase, ointment, suspension, pellets, spray, lozenges, or as an adhesive paste [8]. In the previous studies, the evaluated comparation of the effects of several topical forms of dexamethasone, such as mouthwash, glycerol, mucolox, and pure drug solution, the therapeutic strategies in the treatment of OLP was to a wide spectrum 0.043–25% [7,19,23,24], and within the range the concentration of dexamethasone was 2% in our study. Gentamicin belongs to the aminoglycoside family and is one of the few thermally stable antibiotics, and it can play a synergistic therapeutic role when used in combination with dexamethasone [11]. In this study, the more obvious improvement was at 3 months. The effective rate was 66.7%, similar to previous reports, with comparable local drug efficacy rates [6,19]. However, complications due to corticosteroid treatment in the oral mucosa included refractory cases or relapse of treatment, and development of secondary candidiasis [25], and in an attempt to improve upon this, we aimed to develop an optimized and improved treatment regimen through combination with additional treatment.

TGP are extracted from the dried root of the *Paeonia lactiflora* Pallas, a traditional Chinese medicine that has been used to treat inflammation, pain, and immune disorders for over 1000 years in China [26]. In our study, TGP capsules combined with local administration of the dexamethasone plus gentamicin mouthwash resulted in the total effective rate of 86.4% at the 6-month follow-up. TGP acted as a combination therapy and increased efficacy, with resemblance to the previous study [18]. In that research, the effective rate was 50%, 90.9%, and 100% in the group of dexamethasone acetate 0.1% combined TGP capsules for reticular OLP, while it was 82.4%, 88.2%, and 100% in the group of oral prednisolone combined TGP capsules for erosive OLP at months 1, 3, and 6. All patients rinsed with 1% sodium bicarbonate prophylactically for oral candidiasis. At the same follow-up, the total effective rate in our study was 77.8%, 88.9%, and 88.9% for non-erosive OLP, and 61.5%, 76.9, and 84.6% for erosive OLP in the group of TGP capsules combined with mouthwash. Here, we recruited OLP patients without fungal infection and prophylactic antifungal therapy was not administered. In addition, the effective rate of single site and multiple sites was 71.4% and 93.3% separately at 6 months. The difference of efficacy between both studies may be related to drug type, dosage form, and method of administration. The synergistic effect of TGP may be related to the immunomodulatory and anti-inflammatory functions. Although the pathogenesis of OLP remains unclear, antigen-specific and nonspecific mechanisms include T cell accumulation on the lamina propria surface, basement membrane disruption, intraepithelial T cell migration, and keratinocyte apoptosis [27]. TGP treatment could affect the immunomodulatory function of all responders, as manifested by the continuous reduction of Treg and Th1 numbers [28], and significantly inhibits T cell proliferation [29].

The incidence of oral candidiasis was examined during follow-up, with less than 5% in both groups. A previous study on 315 OLP patients with steroids with or without the use of an antifungal regimen reported that the overall incidence of oral fungal infection was 13.6%, probably due to the fact that previous studies were retrospective, and the baseline assessment of *Candida* spp. growth was lacking [30]. In this study, the enrolled OLP patients were fungus-negative. No prophylactic antifungal therapy was used in OLP treatment in this study, and the percentage of fungal infections that occurred was no higher than the percentage of candida infections in the healthy population. *Candida* spp. are considered a commensal organism in the oral cavity of healthy individuals and could be isolated in asymptomatic people at a rate between 3% and 50% [31]. Unfortunately, in the follow-up, fungal infections of candidiasis were observed both in erosive OLP patients, though with no statistical difference. This was consistent with previous studies showing that OLP patients are more likely to harbor *Candida* spp., especially of erosive or plaque type [32,33]. What is more, the combined administration of TGP did not increase fungal infection. TGP has been widely used to treat autoimmune diseases, with a gentle effect, immune regulation activity, and few adverse drug reactions [26]. During treatment with TGP, no liver and kidney damage or other side effects were observed [34]. A few patients showed diarrhea symptoms [35,36], which quickly dissipated, and only one patient withdrew from our study for this reason.

There were several limitations in the research. First, in order to avoid the fungal infection caused by the long-term use of steroids and antibiotics, the mouthwash regime was chosen to be used for 3 weeks with a 1-week pause. However, a control group should be established to use the same mouthwash continuously in our study. Second, the lack of TGP placebo was the limitation, especially for patients with psychogenic component. If a placebo addition for TGP was established, the evaluation of efficacy may be more definitive. Third, the TGP as monotherapy could be evaluated in future studies. Fourth, it was a convenience sampling in this study and the subjects may be slightly less representative of the patients in the whole country. However, as continuously enrolled, it was a good representation of OLP patients in our hospital. Fifth, the examiner’s calibration data were lacking, as to avoid the pain caused to patients by repeated pulling mucosa during the examination in the study.

## 5. Conclusions

In conclusion, dexamethasone and gentamicin mouthwash has a certain therapeutic effect on OLP without fungal infection, and in combination with TGP capsules can significantly improve the curative effect, especially for erosion-type or more sites of mucosal-involved OLP. According to our study, this approach with integration of traditional Chinese and Western medicine is a safe, effective, and promising strategy for the treatment of OLP.

## Figures and Tables

**Figure 1 jcm-11-07004-f001:**
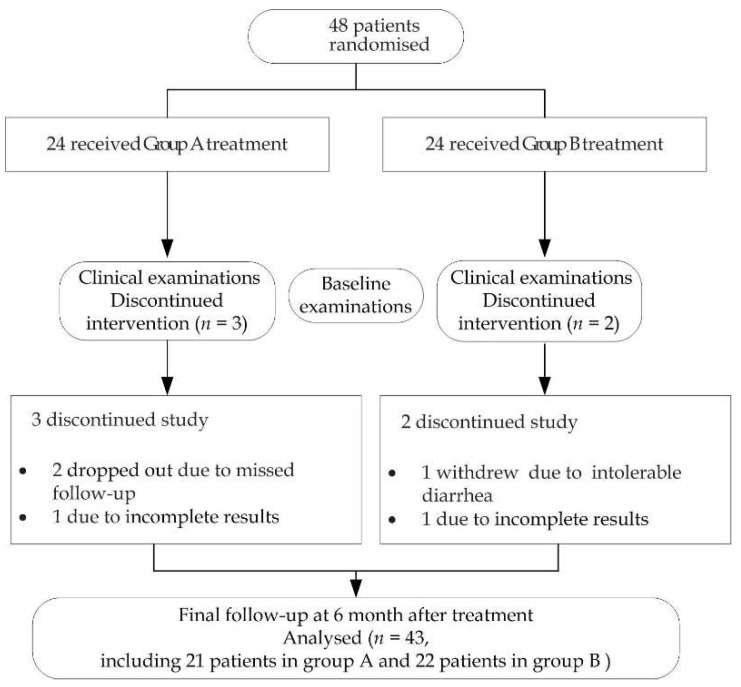
Trial profile. Participants, treatment, and assessment during follow-up.

**Figure 2 jcm-11-07004-f002:**
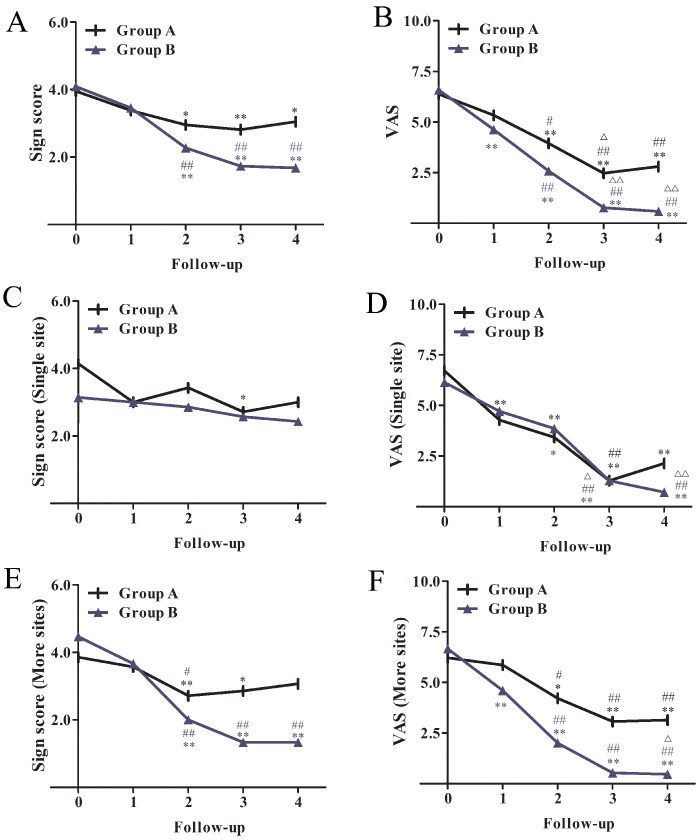
Sign score and VAS trends before and after treatment and during follow-up in each group. (**A**,**C**,**E**) Sign score of all patients (**A**), patients with a single site of mucosal involvement (**C**), patients with more than one site of mucosal involvement (**E**) in Group A and B, (**B**,**D**,**F**) VAS of all patients (**B**), patients with a single site of mucosal involvement (**D**), patients with more than one site of mucosal involvement (**F**) in Group A and B. * Compared with baseline, ** p* < 0.05, *** p* < 0.01, # compared with the 1st follow-up, *# p* < 0.05, *## p* < 0.01, Δ compared with the 2nd follow-up, Δ *p* < 0.05, ΔΔ *p* < 0.01.

**Figure 3 jcm-11-07004-f003:**
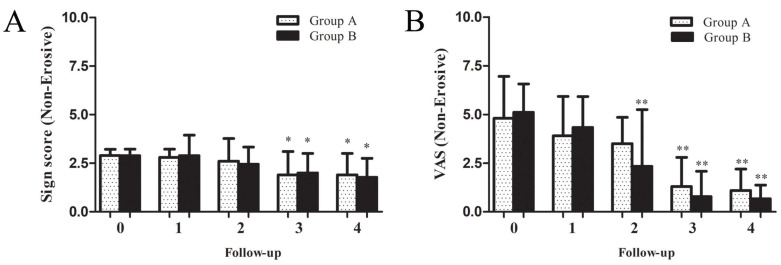
Sign score (**A**) and VAS (**B**) trends before and after treatment of patients with non-erosive OLP in Group A using dexamethasone plus gentamycin mouthwash and Group B by dexamethasone plus gentamycin mouthwash combined with TGP capsules. * Compared with initial visit, * *p* < 0.05, ** *p* < 0.01.

**Figure 4 jcm-11-07004-f004:**
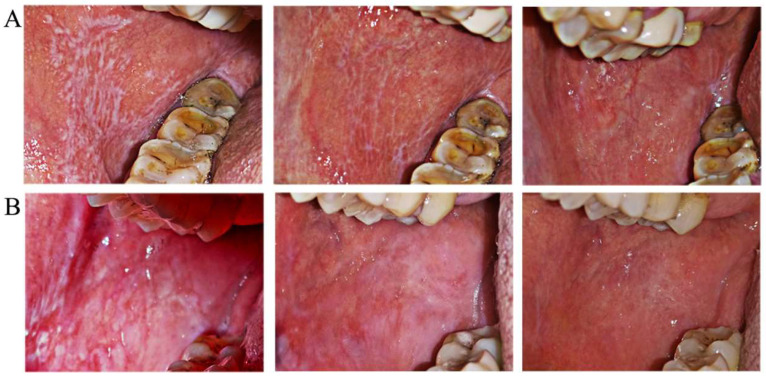
Magnitude of clinical responses of non-erosive OLP to dexamethasone plus gentamycin mouthwash (**A**) and dexamethasone plus gentamycin mouthwash combined with TGP capsules (**B**). The left panel is the clinical presentation at the initial baseline, the middle panel is the clinical presentation at the 1-month follow-up, indicating improved white lesions, and the right panel is the clinical presentation at the 6-month follow-up, when only a few white lesions remained (upper, Group A) or they had disappeared (upper, Group B).

**Figure 5 jcm-11-07004-f005:**
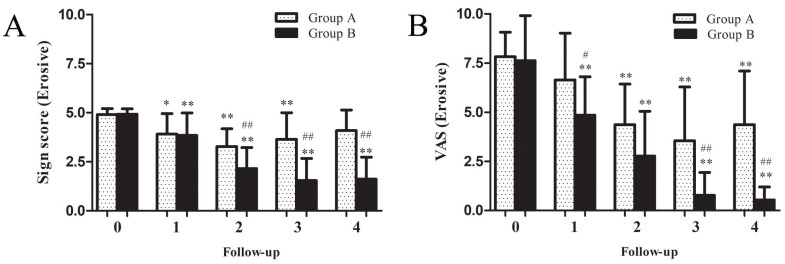
Sign score (**A**) and VAS (**B**) trends before and after treatment of patients with non-erosive OLP in Group A using dexamethasone plus gentamycin mouthwash and Group B by dexamethasone plus gentamycin mouthwash combined with TGP capsules. * Compared with initial visit of the same group, * *p* < 0.05, ** *p* < 0.01, # compared with group A at the same month, # *p* < 0.05, ## *p* < 0.01.

**Figure 6 jcm-11-07004-f006:**
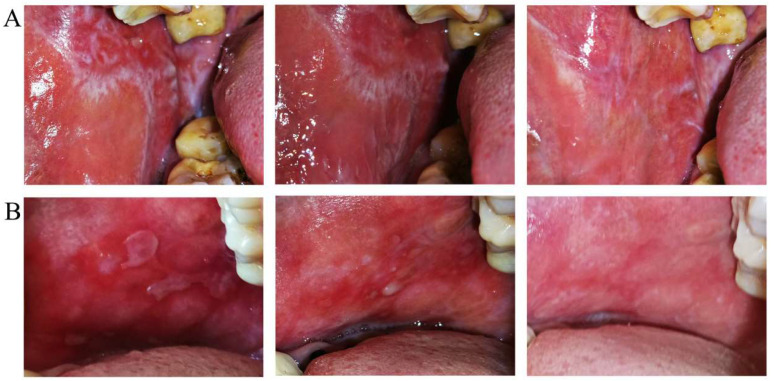
Magnitude of clinical responses of erosive OLP to dexamethasone plus gentamycin mouthwash (**A**) and dexamethasone plus gentamycin mouthwash combined with TGP capsules (**B**). The left panel is the clinical presentation at the initial baseline, the middle panel is the clinical presentation at the 1-month follow-up, indicating improved erosion lesions, and the right panel is the clinical presentation at the 6-month follow-up, showing the erosion had disappeared or a few white lesions remaining (upper, Group A) and erosion had disappeared (upper, Group B).

**Table 1 jcm-11-07004-t001:** General information of patients at baseline (mITT).

	Total	Group A	Group B	*p* Values (A vs. B)
*n*	43	21	22	
Gender				0.586
Male	14 (32.6%)	6 (28.6%)	8 (36.4%)	
Female	29 (67.4%)	15 (71.4%)	14 (63.6%)	
Age range (Years)	49.79 ± 17.52	50.29 ± 18.8	49.32 ± 16.63	0.859
Height	164.98 ± 8.15	164.43 ± 7.71	165.50 ± 8.69	0.672
Weight	62.83 ± 13.16	62.24 ± 15.03	63.38 ± 11.41	0.779
Type				0.658
Non-Erosive	19 (44.2%)	10 (47.6%)	9 (40.9%)	
Erosive	24 (55.0%)	11 (52.4%)	13 (59.1%)	
Sites of mucosal involvement				0.079
Buccal mucosa alone	11 (25.6%)	6 (28.6%)	5 (22.7%)	
Buccal mucosa + gingiva	10 (23.3%)	2 (9.5%)	8 (36.4%)	
Buccal mucosa + tongue	9 (20.9%)	8 (38.1%)	1 (4.5%)	
Buccal mucosa + palate	4 (9.3%)	2 (9.5%)	2 (9.1%)	
Buccal mucosa + lip	4 (9.3%)	2 (9.5%)	2 (9.1%)	
Buccal mucosa + gingiva + tongue + lip	1 (2.3%)	0 (0.0%)	1 (4.5%)	
Gingiva alone	2 (4.7%)	0 (0.0%)	2 (9.1%)	
Gingiva + tongue	1 (2.3%)	0 (0.0%)	1 (4.5%)	
Palate alone	1 (2.3%)	1 (4.8%)	0 (0.0%)	

**Table 2 jcm-11-07004-t002:** Clinical response (sign score) and symptom score (VAS) at follow-up assessment by treatment (mITT).

		Sign Score			VAS	
	Group A (*n* = 21)	Group B (*n* = 22)	*p* Values	Group A (*n* = 21)	Group B (*n* = 22)	*p* Values
Baseline	3.95 ± 1.07	4.05 ± 1.13	0.795	6.38 ± 2.29	6.59 ± 2.32	0.745
First follow-up	3.38 ± 0.97	3.45 ± 1.18	0.838	5.33 ± 2.58	4.64 ± 1.79	0.281
Second follow-up	2.95 ± 1.07	2.27 ± 0.98	0.059	3.95 ± 1.77	2.59 ± 2.50	0.036
Third follow-up	2.81 ± 1.54	1.73 ± 1.08	0.003	2.48 ± 2.46	0.77 ± 1.19	0.009
Fourth follow-up	3.05 ± 1.53	1.68 ± 1.04	<0.001	2.81 ± 2.66	0.59 ± 0.66	0.001

Group A: Treated with dexamethasone plus gentamycin mouthwash. Group B: Treated with dexamethasone plus gentamycin mouthwash combined with TGP capsules. Mean ± SD.

**Table 3 jcm-11-07004-t003:** Comparison of the treatment effective rate between the two groups (mITT).

	Group	Course of Disease (Follow-Up)			
	(A *n* = 21, B *n* = 22)	1	2	3	4
Complete resolution	A	0 (0.0%)	0 (0.0%)	1 (4.8%)	1 (4.8%)
	B	0 (0%)	1 (4.5%)	4 (18.2%)	4 (18.2%)
Partial resolution	A	7 (33.3%)	11 (52.4%)	13 (61.9%)	12 (57.1%)
	B	10 (45.5%)	14 (63.6%)	14 (63.6%)	15 (68.2%)
No resolution	A	14 (66.7%)	10 (47.6%)	7 (33.3%)	8 (38.1%)
	B	12 (54.5%)	7 (31.8%)	4 (18.2%)	3 (13.6%)
Total effective rate	A	7 (33.3%)	11 (52.4%)	14 (66.7%)	13 (61.9%)
	B	10 (45.5%)	15 (68.2%)	18 (81.8%)	19 (86.4%)
*p* Values					0.007

**Table 4 jcm-11-07004-t004:** Comparison of sign score and VAS between groups with different sites of mucosal involvement.

Sign Score	Single Site of Mucosal Involvement			More Sites of Mucosal Involvement		
	Group A (*n* = 7)	Group B (*n* = 7)	*p* Values	Group A (*n* = 14)	Group B (*n* = 15)	*p* Values
Baseline	4.14 ± 1.07	3.14 ± 1.07	0.101	3.86 ± 1.10	4.47 ± 0.92	0.150
First follow-up	3.00 ± 0.00	3.00 ± 1.15	1.000	3.57 ± 1.16	3.67 ± 1.18	0.822
Second follow-up	3.43 ± 1.13	2.86 ± 0.38	0.348	2.71 ± 0.99	2.00 ± 1.07	0.092
Third follow-up	2.71 ± 1.89	2.57 ± 0.79	0.814	2.86 ± 1.41	1.33 ± 0.98	<0.001
Fourth follow-up	3.00 ± 1.53	2.43 ± 0.79	0.348	3.07 ± 1.59	1.33 ± 0.98	<0.001
**VAS**						
Baseline	6.71 ± 1.50	6.14 ± 1.86	0.609	6.21 ± 2.64	6.67 ± 2.41	0.560
First follow-up	4.29 ± 1.60	4.71 ± 1.60	0.701	5.86 ± 2.85	4.60 ± 1.92	0.106
Second follow-up	3.43 ± 1.13	3.86 ± 2.85	0.701	4.21 ± 2.01	2.00 ± 2.17	0.005
Third follow-up	1.29 ± 1.25	1.29 ± 1.25	1.000	0.07 ± 2.73	0.53 ± 1.13	0.001
Fourth follow-up	2.14 ± 2.48	0.71 ± 0.49	0.202	3.14 ± 2.77	0.47 ± 0.74	0.001

**Table 5 jcm-11-07004-t005:** Comparison of the treatment effective rate between groups with different sites of mucosal involvement.

	Single Site of Mucosal Involvement (Group A *n* = 7, Group B *n* = 7)				More Sites of Mucosal Involvement (Group A *n* = 14, Group B *n* = 15)			
**Follow-Up**	**1**	**2**	**3**	**4**	**1**	**2**	**3**	**4**
Group A								
Complete resolution	0 (0.0%)	0 (0.0%)	1 (14.3%)	(0.0%)	0 (0.0%)	0 (0.0%)	0 (0.0%)	1 (7.1%)
Partial resolution	3 (42.9%)	5 (71.4%)	4 (57.1%)	5 (71.4%)	4 (28.6%)	6 (42.9%)	9 (64.3%)	7 (50.0%)
No resolution	4 (57.1%)	2 (28.6%)	2 (28.6%)	2 (28.6%)	10 (71.4%)	8 (57.1%)	5 (35.7%)	6 (42.9%)
Effective rate	3 (42.9%)	5 (71.4%)	5 (71.4%)	5 (71.4%)	4 (28.6%)	6 (42.9%)	9 (64.3%)	8 (57.1%)
Group B								
Complete resolution	0 (0.0%)	0 (0.0%)	0 (0.0%)	0 (0.0%)	0 (0%)	1 (6.7%)	4 (26.7%)	4 (26.7%)
Partial resolution	2 (28.6%)	3 (42.9%)	6 (85.7%)	5 (71.4%)	8 (53.3%)	11 (73.3%)	8 (53.3%)	10 (66.7%)
No resolution	5 (71.4%)	4 (57.1%)	1 (14.3%)	2 (28.6%)	7 (46.7%)	3 (20.0%)	3 (20.0%)	1 (6.7%)
Total effective rate	2 (28.6%)	3 (42.9%)	6 (85.7%)	5 (71.4%)	8 (53.3%)	12 (80.0%)	12 (80.0%)	14 (93.3%)
*p* Values				0.376				0.003

**Table 6 jcm-11-07004-t006:** Comparison of treatment effective rate between two groups with non-erosive and erosive OLP.

	Non-Erosive OLP (Group A *n* = 10, Group B *n* = 9)				Erosive OLP (Group A *n* = 11, Group B *n* = 13)			
**Follow-Up**	**1**	**2**	**3**	**4**	**1**	**2**	**3**	**4**
Group A								
Complete resolution	0 (0.0%)	0 (0.0%)	1 (10.0%)	1 (10.0%)	0 (0.0%)	0 (0.0%)	0 (0.0%)	0 (0.0%)
Partial resolution	5 (50.0%)	7 (70.0%)	7 (70.0%)	7 (70.0%)	2 (18.2%)	4 (36.4%)	6 (54.5%)	5 (45.5%)
No resolution	5 (50.0%)	3 (30.0%)	2 (20.0%)	2 (20.0%)	9 (81.8%)	7 (63.6%)	5 (45.5%)	6 (54.5%)
Effective rate	5 (50.0%)	7 (70.0%)	8 (80.0%)	8 (80.0%)	2 (18.2%)	4 (36.4%)	6 (54.5%)	5 (45.5%)
Group B								
Complete resolution	0 (0%)	0 (0.0%)	1 (11.1%)	1 (11.1%)	0 (0.0%)	1 (7.7%)	3 (23.1%)	3 (23.1%)
Partial resolution	4 (44.4%)	7 (77.8%)	7 (77.8%)	7 (77.8%)	6 (46.2%)	7 (53.8%)	7(53.8%)	8 (61.5%)
No resolution	5 (55.6%)	2 (22.2%)	1 (11.1%)	1 (11.1%)	7 (53.8%)	5 (38.5%)	3 (23.1%)	2 (15.4%)
Effective rate	4 (44.4%)	7 (77.8%)	8 (88.9%)	8 (88.9%)	6 (46.2%)	8 (61.5%)	10 (76.9%)	11 (84.6%)
*p* Values				0.582				0.037

**Table 7 jcm-11-07004-t007:** Adverse events (AEs) in both groups with certain or possible causal relationships to the applied drugs (SS).

AEs	Group	Course of Disease (Follow-Up)			
		1	2	3	4
Oral AE					
Fungal infection (candidiasis)	A	0	1 (4.2%)	0	0
	B	0	0	1 (4.2%)	0
Systemic AE					
Gastrointestinal symptoms (diarrhea)	A	0	0	0	0
	B	1 (4.2%)	0	0	0
Liver function (total bilirubin)	A	0	0	0	0
	B	0	0	0	1 (4.2%)
*p* Values					0.554

## Data Availability

The data are obtainable on request from the corresponding author in this study. They are not publicly available as privacy issues.

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
