# Peer review of "Clinical Evaluation of Dexamethasone Plus Gentamycin Mouthwash Use in Combination with Total Glucosides of Paeony for Treatment of Oral Lichen Planus without Fungal Infection: A Comparative Study with Long-Term Follow-Up"

_jcm, 2022, doi:10.3390/jcm11237004_

Round 1
Reviewer 1 Report
Minor concerns:
- English language (please check spelling, and grammar).
Abstract:
- Define the total effective rate!
- Avoid conclusions not related to the study, e.g. claims about prophylactic treatment with antifungal drugs!
Introduction:
- May be shortened by half with a more concise writing style.
- Elaborate on the use of gentamicin for the treatment of OLP. In particular, add quotations to relevant clinical studies describing the effectiveness of its use.
- Mention adhesive media for local delivery; explain why you decided to use solutions and avoid adhesive media.
- Elaborate on the chemical composition of TGP.
- Line 74: please provide a reference for ‘Local application of gentamicin injection in combination with dexamethasone may be a promising approach for the inflammation of OLP’. Why was this exact combination of drugs chosen as one of the treatment options?
- Please define the objectives of the study more clearly and concisely. What were the outcome measures?
- Please use the abbreviation TGP consistently.
MM:
- It is stated that the study was designed as a randomized controlled clinical trial – please make sure that the paper includes all the information required by the CONSORT checklist.
- Line 129: please add the city and country of the hospitals, at which the study was performed.
- How were the patients chosen? Were they selected out of a pool of consecutively evaluated individuals?
- Line 146: as stated, dexamethasone and gentamicin were mixed separately with sodium chloride; were they later combined into one mouthwash solution, or did the patients rinse their oral cavities with separate solutions? If so, what was the sequence of use and why?
- Line 148: why/based on what was the mouthwash regime chosen (3 weeks of use + 1-week pause)? Please clarify and add a reference.
- Line 151: for how long did patients in group B receive the TGP capsules – for the same time period as the mouthwash (until the symptoms subsided)?
- How was compliance control checked?
- Line 156: please elaborate on the oral health guidance and oral hygiene instructions – what did this include?
- Please clearly identify the pre-specified primary and secondary outcome measures.- description of the dental and periodontal status of patients is missing.
- The concentration of dexamethasone used must be supported by quotation.
- Why was a placebo not used? First, it could be done without much difficulty. Second, it would enable allocation concealment. It is a crucial aspect, as OLP is known to have a robust psychogenic component. Therefore, the lack of placebo has to be commented on in the discussion as a severe study limitation.
- Why not use TGP as monotherapy?
- It is necessary to carry out a calibration exercise and define the reproducibility of the “sign score”, as well as the categories “complete resolution”, “partial resolution” and “no resolution”. These are subjective assessments, so it is necessary to determine their reliability. It is also necessary to state who the evaluator was and whether he/she was a different person than the person who administered the treatment.
- How was blinding ensured?
- Is beta = 0.2 the right value for this parameter? Usually, this value is set at 0.8. A beta of 0.2 would mean that the statistical power is deficient, meaning there is a high probability of inconclusive results.
- The entire statistical analysis is questionable. Outcomes are categorical rather than continuous numerical variables, requiring appropriate statistical tests. E.g. calculating the mean and standard deviation from the sign score is wrong. It is also incorrect for VAS, as the linearity between the units of the VAS scale is questionable.
The results
-
-- Please add more demographic information regarding the patients – what other systemic diseases did they suffer from concurrently? Were they taking other regular medication, which could possibly cause drug-induced OLP (if not, please add this to the inclusion/exclusion criteria)?
- For how long (on average) did the patients use the prescribed treatment regime (line 148: ‘until symptoms subsided’)?
- Due to incorrect statistical analysis, the way the results are presented is wrong.
- The VAS charts should have a scale of up to 10 to show that only mild pain was present clearly.
Discussion
- In the first paragraph, summarize the main result of the study.
- Why did you not use TGP as monotherapy? Finally, you don't know if it's a TGP effect, an interaction between all three drugs, or maybe just a placebo effect. Please add this as a research limitation.
- The discussion should be shortened by half and include only data relevant to understanding the research results.
Reviewer 2 Report
Dear Authors,
Congratulation for your clinical study. Any inovative treatment approach for OLP is important, as it is considered a potentially malignant lesion. I found the Introduction section very long (lines 87-117 could be removed or placed in Discussion section) and some of the information are repeated in the Discussion.
I would appreciate if you could explain more clear the aim of the study.
You could provide more references: in lines 44-49 where you mention factors important in the etiology of OLP, lines 76-82, lines 84-86 where you report that you formulated a mouthwash and I wonder if you have some published data regarding this product or its use. Out of 52 references, more than half are in Introduction, but Discussions are 3 times longer. I could not find references 33,34.
For Material and Methods, I understand that only one person made all examinations during the study, otherwise you should give a kappa coefficient for calibration among examiners. Who completed the VAS score? What was the patient contribution to the evaluation of pain intensity?
Why didn`t you mention a clinical study closely related to your investigation? (Zhou L, Cao T, Wang Y, Yao H, Du G, Tian Z, Tang G. Clinical observation on the treatment of oral lichen planus with total glucosides of paeony capsule combined with corticosteroids. Int Immunopharmacol. 2016 Jul;36:106-110. doi: 10.1016/j.intimp.2016.03.035. Epub 2016 Apr 26. PMID: 27129091.)
I think the conclussion are too optimistic, I would expect to ask for further clinical studies to confirm your results and also to discuss weather this study has some limitations.
Round 2
Reviewer 1 Report
Thank you for considering most of the comments. However, some points remain unexplained.
Objectives (point 8) are confusing, and the paragraph describing them needs serious English editing. Then, based on the research design, a hypothesis could be formulated.
In point 12, the method of sampling is still not explained. Was it convenience sampling? If yes, this should be considered a substantial limitation.
The question in point 17 was not understood correctly. Did you evaluate oral hygiene with any plaque index?
The answer to point 19 needs to be supported with several references (not just one).
The answer to point 22 still lacks data on the examiner’s calibration (as measures of accuracy and repeatability).
The charts presented in Figs 3 and 5 must have the same scale or range on the y-axis. Otherwise, they are misleading.
Overall, professional proofreading is strongly advised.
